# Multifunctional Partially Reflective Surface for Smart Blocks

**DOI:** 10.3390/s21196508

**Published:** 2021-09-29

**Authors:** Jae Hee Kim, Dong-Jin Lee, Tae-Ki An, Jong-Gyu Hwang, Chi-Hyung Ahn

**Affiliations:** 1School of Electrical, Electronics and Communication Engineering, Korea University of Technology and Education, Cheonan-si 31253, Chungcheongnam-do, Korea; jaehee@koreatech.ac.kr; 2Train Control and Communications Research Department, Korea Railroad Research Institute, Uiwang-si 16105, Gyeonggi-do, Korea; dong905@krri.re.kr (D.-J.L.); jghwang@krri.re.kr (J.-G.H.); 3New Transportation Innovative Research Center, Korea Railroad Research Institute, Uiwang-si 16105, Gyeonggi-do, Korea; tkahn@krri.re.kr

**Keywords:** smart block, Fabry-Perot cavity antenna, partially reflective surface (PRS), gain, polarization

## Abstract

In general, a partially reflective surface (PRS) is mainly used to increase the gain of an antenna; some metallic objects placed on the PRS degrades the antenna performance because the objects change the periodic structure of the PRS. Herein, we propose a multifunctional PRS for smart block application. When a passenger passes over a smart block, the fare can be simultaneously collected and presented through the LED display. This requires high gain antenna with LED structure. The high gain characteristic helps the antenna identify passengers only when they pass over the block. The multifunctional PRS has a structure in which an LED can be placed in the horizontal direction while increasing the antenna gain. We used the antenna’s polarization characteristics to prevent performance deterioration when LED lines are placed in the PRS. We built the proposed antenna and measured its performance: At 2.41 GHz, the efficiency was 81.4%, and the antenna gain was 18.3 dBi. Furthermore, the half-power beamwidth was 18°, confirming a directional radiation pattern.

## 1. Introduction

Smart blocks are devices that include various sensing functions in walking paths to improve convenience. In recent years, research has been conducted on smart blocks to improve passenger convenience in railway stations [1]. A typical use case of smart blocks is position detection. If position detection is used in a railway station, information regarding paths to various amenities can be provided [2]. Another use of smart blocks is to collect the fare and notify the system when a person passes the smart blocks. As such, fare collection as people pass over the blocks reduces congestion in the station during morning and evening rush hours because separate entrance gates are no longer needed.

Figure 1 shows the proposed smart block system structure. An Internet of Things (IoT) device is embedded in a block, and light-emitting diodes (LEDs) are installed on certain areas on the surface of the block. When a user passes over a smart block, the fare is collected through communication, and the collected fare is indicated using the LED display on the floor. As the smart block must execute its function when someone passes over it, the IoT device should be designed with directional electromagnetic waves. It is also important that the LED elements installed on the surface should not adversely affect the performance of the antenna.

Many methods have been proposed for designing directional antennas. Typically, an antenna array is used [3,4,5,6]. This method arranges multiple antennas that are approximately a half-wavelength apart and excite the same amplitude of the signal at each antenna with the same phase. If the number of antennas is increased in the antenna array, the gain of the antennas increases. However, the drawback of this approach is that as the number of antennas increases, the structure of the feed network becomes more complex, causing significant losses. When applying an antenna array to a smart block, it must be located at the lower part of the block to prevent damage to the antennas. Due to this consideration, various types of materials over the antennas and electronic devices such as LEDs may decrease antenna efficiency and distort radiation patterns Therefore, it is necessary to design an antenna structure more suitable for smart blocks.

Recently, methods have been proposed to increase the antenna gain while overcoming the antenna array complexity, in which a dielectric structure or metamaterial is placed on a single-patch antenna or the upper surface. In the case of using a dielectric, a method has been proposed that uses microstrip antennas stacked with dielectric layers with different permittivities to increase the directivity [7,8,9,10]. When this method is used, the structure is complex because multiple dielectric layers are stacked. Alternatively, the directivity can be increased by drilling a hole on the dielectric, thereby changing the effective permittivity value. However, this design is difficult to realize because the mechanical shape of the dielectric must be changed.

Many studies have been conducted on increasing the directivity by using metamaterials. In general, a metamaterial plane is inserted at the top of the microstrip single-patch antenna [11,12,13,14,15]. This approach is based on the principle that the metamaterial reflects a portion of the electromagnetic waves radiated from the inside of the patch antenna, which is then reflected again by the ground and reaches the antenna, thus increasing the radiating aperture area. This results in an increase in the antenna gain. An antenna using a metamaterial in such a manner is often referred to as a Fabry-Perot cavity antenna. Furthermore, because the metamaterial reflects a portion of the electromagnetic waves, the metamaterial plane is called a partially reflective surface (PRS). The gain of a Fabry-Perot cavity antenna is determined by the shape and size of the surface located at the top of the antenna and by the gap between the antenna and ground conductor. In particular, the distance between the ground and the upper plate is directly related to the frequency for increasing the directivity. The Fabry-Perot cavity antenna can be easily built because its structure is simple. Moreover, it is highly efficient because it contains no feed network.

Research on PRS has been actively conducted in recent years, from the equivalent circuit model to the beam steering technique [16,17,18]. However, there are few cases of designing PRS shape considering polarization. In the previous research, PRS is used to electronically alter antenna polarization between linear polarization and circular polarization using PIN diodes [19]. This paper, to apply PRS to a smart block, presents a PRS structure that can mount LED lines while increasing antenna gain using polarization characteristic. In Section 2, we describe the unit cell simulations used to design a PRS for which the addition of LEDs does not affect the antenna performance. In Section 3, we discuss the fabrication of the entire antenna structure and the comparison of the performance measurement results with those of the simulations. Finally, Section 4 presents the conclusions.

## 2. Design of the Antenna

Figure 2 shows the structure of the proposed antenna. The patch antenna is at the bottom, and the PRS is at the top. We implemented the antenna using three FR4 substrates with a permittivity of 4.4 and a thickness of 1 mm. The first substrate was used as the ground, which had horizontal and vertical lengths of 400 mm and 400 mm, respectively. A second substrate was used to fabricate the patch antenna. The horizontal and vertical lengths of the patch were 38 and 39 mm, respectively. In general, the length of a patch antenna is determined based on the half-wavelength of the design frequency. When designing at 2.4 GHz, the length should be set to approximately 43 mm, considering the permittivity. However, when a PRS is placed on top of the patch antenna, the resonance frequency of the patch antenna is distorted; thus, the length of the antenna was set to 39 mm in this study, considering impedance matching. The feed of the patch antenna was positioned 12 mm below the center of the patch.

The final substrate was used to design the PRS. The size of the PRS was the same as that of the ground: 400 mm horizontally and 400 mm vertically. The most important factor in designing a PRS to increase directivity is the distance between the ground and the PRS. The height from the ground was set to 64 mm, which corresponded to approximately a half-wavelength; this distance provides the maximum directivity at 2.4 GHz. Based on analytical models, such as the transmission line model, the distance between the ground plane and the PRS is the distance which causes the phase sum of the reflected waves to become zero, which is about a half wavelength [16,20].

The function of the PRS is to increase the antenna directivity despite the presence of LED lines consisting of metals. We used the polarization characteristics of the electromagnetic waves to minimize the effect of the LED lines. Figure 3 shows the boundary conditions of the electromagnetic waves. An electric field is formed in the x-direction, and a magnetic field is formed in the y-direction. The propagation direction of the electromagnetic waves is in the z-direction. According to the boundary condition of the electromagnetic waves, if there are periodic metallic lines in the same direction as the electric field, the electric field on the conductor surface becomes zero; consequently, the electromagnetic waves cannot penetrate it and are reflected. On the other hand, if there are periodic metallic lines in the direction perpendicular to the direction of the electric field, the electric field remains as-is without being removed and passes through the conductor. In general, most antennas exhibit linear polarization characteristics. Therefore, if the polarization direction of the antenna is designed to be perpendicular to the surrounding metal object, it will be virtually unaffected by the surrounding metal. As such, the metal structure can be constructed perpendicular to the polarization, or the metal structure can be used for other purposes, such as increasing the mechanical strength.

We performed unit cell simulations to design the PRS, considering the polarization characteristics of the antenna. For the simulations, we used CST MWS, a commercial electromagnetic simulation tool. In general, the performance of the PRS is estimated through unit cell simulations, and the shape or size of a unit cell determines the primary PRS characteristics. Figure 4 shows the structure of the unit cell simulation, in which a rectangular patch was used for the basic shape of the PRS. The size of the rectangular patch was 55 mm, both horizontally and vertically. The horizontal and vertical lengths of each unit cell were 60 mm. This means that the unit cells were placed at intervals of 60 mm in the PRS when the actual PRS was constructed. The length of 60 mm was selected because it is the distance corresponding to approximately a half-wavelength at 2.4 GHz. To form the electric field in the x-axis direction, the boundary of the unit cells was set with a perfect electric conductor (PEC) at the upper and lower surfaces and with a perfect magnetic conductor (PMC) on the left and right sides.

For the PRS, the waves reflected by the ground are transmitted to the PRS again. To consider the effect of the ground, a mirror image of the PRS should be created when simulating the unit cells. This requires configuring it to be exactly symmetrical to the existing PRS. As a mirror image of the PRS has been created, the metal ground surface must be removed. However, the material used for the ground surface was left intact in our simulation, to consider the effect of the dielectric. We examined whether the LED lines affected the performance of the PRS when conducting unit cell simulations. Therefore, when conducting unit cell simulations, we created metal lines horizontally and set the width to *w*. We checked whether the simulation results of the unit cells varied depending on the width of the metal lines.

Figure 5 shows the unit cell simulation results. We conducted simulations for cases where the metal line widths were 0 mm, 10 mm, and 20 mm. In the case of the metal line width *w* = 10 mm, the simulation results were identical to those of the case of *w* = 0 mm (a situation where there was no metal line). Even when the width was increased to 20 mm, the results were nearly the same, although the frequency moved slightly. In other words, if the polarization direction of the antenna is positioned perpendicular to the LED lines, the transmission and reflection characteristics of electromagnetic waves are nearly unaffected. This means that there is no effect, even if the LEDs are attached to the PRS surface.

Figure 6 illustrates the structure of the final PRS design. The PRS was patterned with metal on an FR4 substrate with a thickness of 1 mm. Five unit cells were placed horizontally, and five more were placed vertically. In the case of horizontal metal lines on which LED lines can be attached, the width was set to 10 mm. This value is sufficiently large to attach an actual LED line because the width of the latter is less than 10 mm.

## 3. Results of Simulations and Experiments

The complete structure of the proposed antenna is shown in Figure 7a,b shows the parts excluding the PRS at the top. Three FR4 substrates were used to build the antenna. The bottom side is FR4, which has only the ground; the second FR4 is a substrate with only the patch; and the third is the PRS, which increases the directivity without being affected by the LEDs. There was a gap between the ground and patch substrate, with an air layer of 3 mm. To support this structure, dielectric posts were placed at the four corners of the substrates where the antenna was not positioned. A coaxial cable was used to feed the antenna. The center conductor was soldered to the patch, and the outer conductors were soldered to the ground side. Eight columns were used to maintain a constant gap between the PRS and the ground. To examine the operating frequency of the constructed antenna, we measured |S_11_| using a network analyzer. First, we measured |S_11_| for the patch antenna, excluding the PRS, as illustrated in Figure 8. In the case of the patch antenna without a PRS, the resonance frequency was formed at 2.74 GHz, and the simulation results were quite consistent with the experimental results. Figure 9 shows the simulated and measured |S_11_| for the case of including the PRS.

Even with the presence of a PRS, the simulation and measurement results were similar, and resonance was formed at 2.4 GHz. In the simulations, the impedance bandwidth of the antenna with |S_11_| < −10 dB was 2.393–2.423 GHz (30 MHz); the measurement result of the constructed antenna was 2.407–2.437 GHz (30 MHz). The measurement results showed very slightly higher resonance frequencies, which seemed to be due to a slight variation in certain parameters, such as the difference in permittivity between the simulated and actual substrates.

We measured the antenna efficiency and gain in an anechoic chamber with dimensions of 15.2 × 7.9 × 7.9 m, as shown in Figure 10. Figure 11 shows a comparison of the measurement and simulation results for the antenna gain and efficiency. The measured values were lower than the simulation results. For all frequency bandwidths, the simulation and measurement results showed similar trends. The efficiency was slightly lower in the measurement results because of the additional loss caused by the feed cable. The frequency that produced the maximum efficiency was 2.41 GHz in both simulation and measurement; the maximum efficiency was 93.2% in the simulation results and 81.4% in the measurement results. The maximum gain also occurred at 2.41 GHz; the corresponding values were 19.2 dBi in the simulation and 18.3 dBi in the experiment.

Figure 12 shows the simulated and measured radiation patterns, including the power sum for the E-plane and H-plane described in Figure 10b. In general, if a PRS is used to increase the antenna gain, the bandwidth that maintains a high gain is reduced as the antenna gain increases. We selected 2.36, 2.41, and 2.46 GHz as the measurement frequencies to examine the extent to which the radiation pattern is distorted when the frequency varies slightly from the design frequency. The comparison shows that the simulation and measurement results are fairly similar at each frequency for the E- and H-planes. Furthermore, even if the frequency is approximately 50 MHz from the design frequency, the radiation pattern is not distorted significantly. The measured gain was 11.6 dBi at 2.36 GHz, 18.3 dBi at 2.41 GHz, and 16.7 dBi at 2.46 GHz, which indicates that the gain drops more at the side below the design frequency.

It is important in the train station application for a smart block to recognize a person only when he or she walks over it. Therefore, it is also important to check the 3-dB beamwidth of the radiation pattern. Table 1 shows the 3-dB beamwidth, which is 18° at the design frequency of 2.41 GHz; the simulation and measurement results are quite similar. Therefore, the beam width is sufficiently narrow, and is suitable for recognizing people passing over the smart blocks.

## 4. Conclusions

In this study, we designed a multifunctional PRS for use in smart blocks. The smart blocks should recognize people passing above, and the recognition result should be indicated using the LED. This requires designing an antenna with high directivity, even in the presence of LEDs. The PRS structure is generally known to increase the directivity, but a metal object such as an LED placed on the PRS can cause significant performance deterioration. Considering the polarization of the antenna when designing the PRS, we designed a structure that is not affected, even if the LED lines are inserted horizontally. The structure of inserting lines in the horizontal direction was combined with the conventional PRS structure, and the unit cell simulations showed that the lines in the horizontal direction had virtually no performance effect. The designed antenna was simulated and constructed. In the actual measurement results of the constructed antenna, the gain was 18.3 dBi at 2.41 GHz, and the 3-dB beamwidth was 18°. The measured efficiency was 81%. If the multifunctional PRS designed in this study is used, it allows LEDs to be easily inserted into the surface of walking blocks. Furthermore, because it is not necessary to design an antenna array to increase the directivity, a feed network need not be designed; this eliminating the loss caused by the feed network. The proposed structure can be implemented with smart blocks provided with embedded Internet of Things (IoT) sensors. In a real-world context, it can be used for automated train fare collection; when a passenger passes over a smart block on a train station platform, the fare can be simultaneously collected and shown on the LED display. Due to its highly directed nature, the antenna can identify passengers only when they pass over the block.

## Figures and Tables

**Figure 1 sensors-21-06508-f001:**
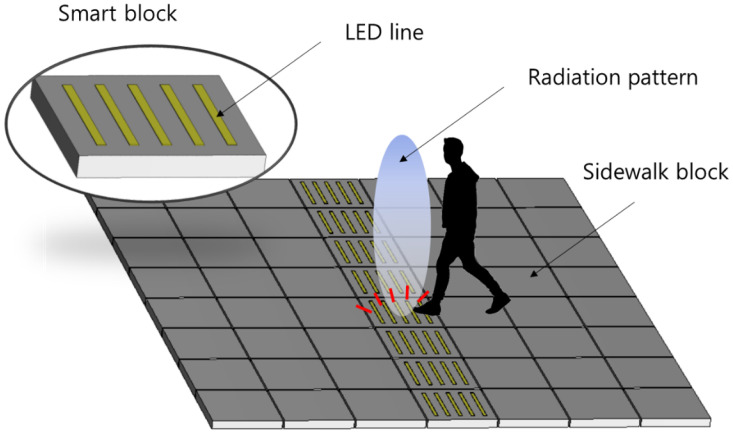
Schematic diagram of smart blocks for fare collection.

**Figure 2 sensors-21-06508-f002:**
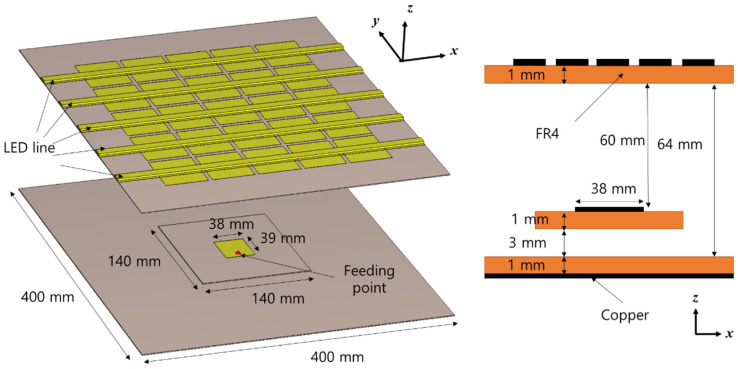
Structure of the proposed antenna.

**Figure 3 sensors-21-06508-f003:**
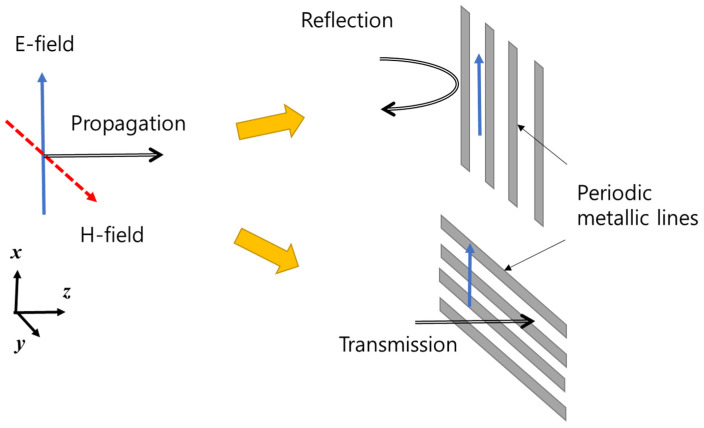
Boundary conditions of electromagnetic waves.

**Figure 4 sensors-21-06508-f004:**
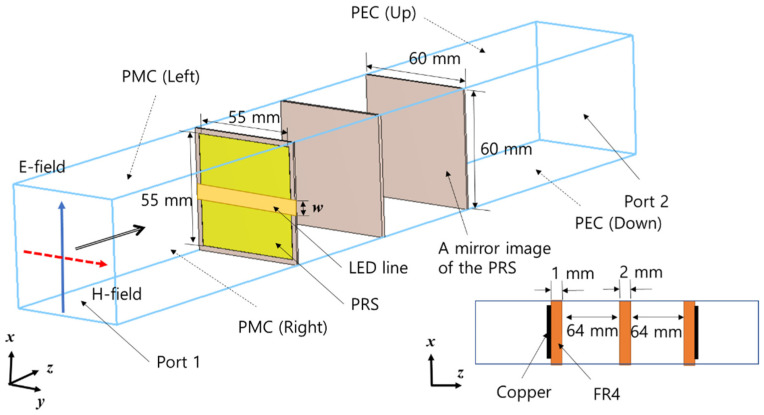
Unit cell simulation structure.

**Figure 5 sensors-21-06508-f005:**
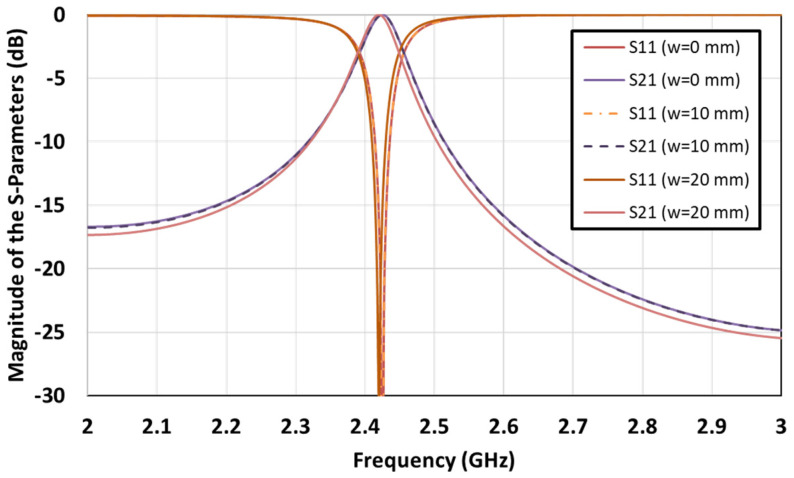
Magnitude of the reflection (S_11_) and transmission (S_21_) coefficients according to the width (*w*) of the LED lines.

**Figure 6 sensors-21-06508-f006:**
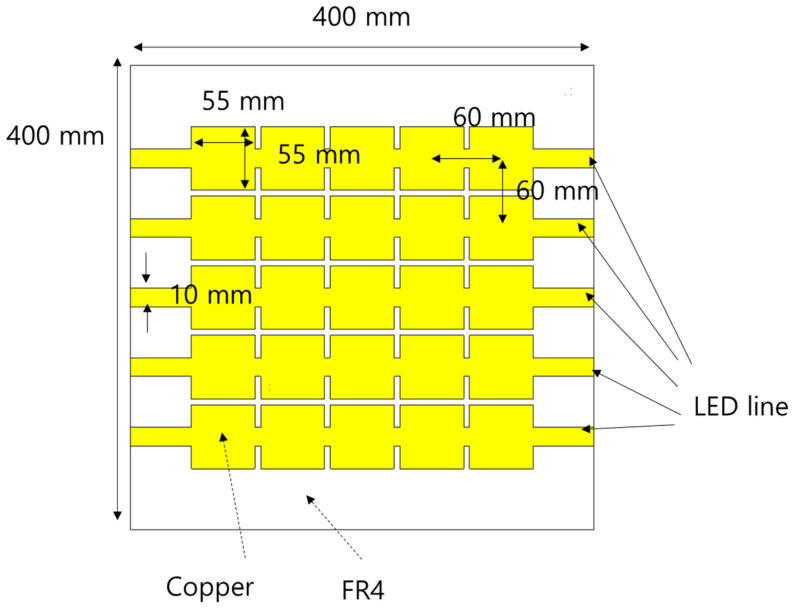
Top view of the proposed PRS.

**Figure 7 sensors-21-06508-f007:**
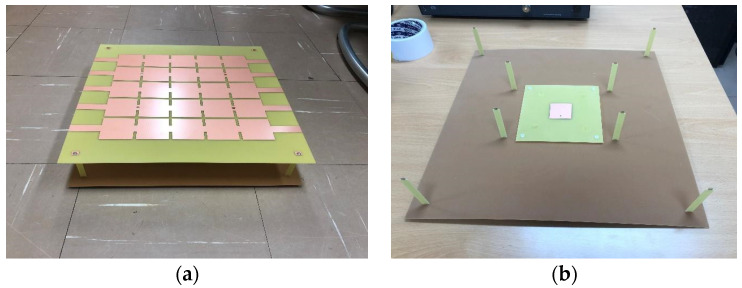
Constructed antenna: (**a**) complete structure, (**b**) patch antenna for feed.

**Figure 8 sensors-21-06508-f008:**
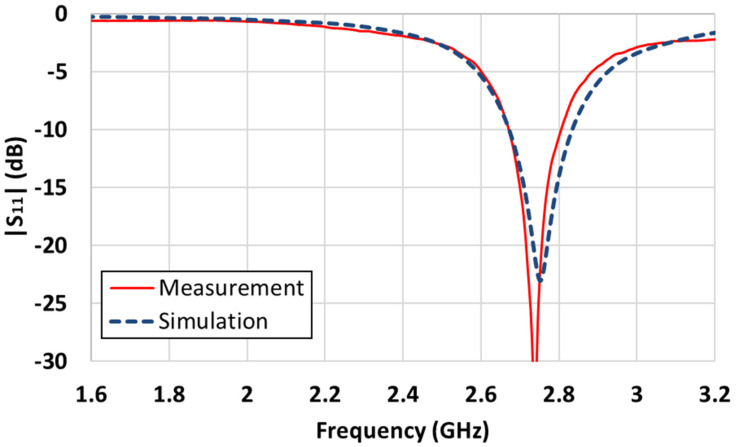
Simulated and measured |S_11_| of the patch antenna excluding a PRS.

**Figure 9 sensors-21-06508-f009:**
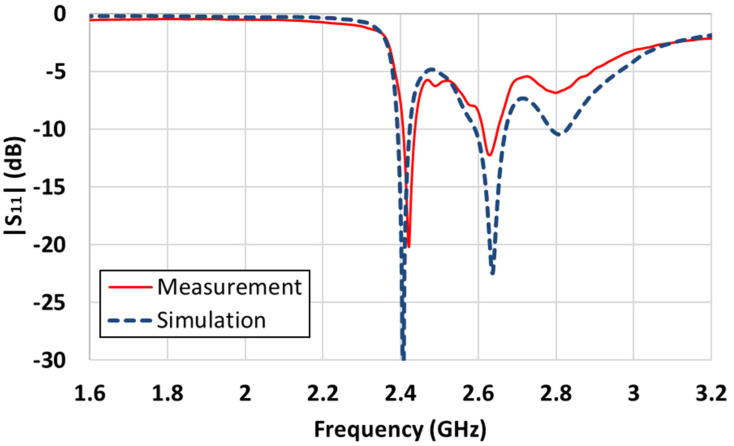
Simulated and measured |S_11_| of a Fabry-Perot cavity antenna.

**Figure 10 sensors-21-06508-f010:**
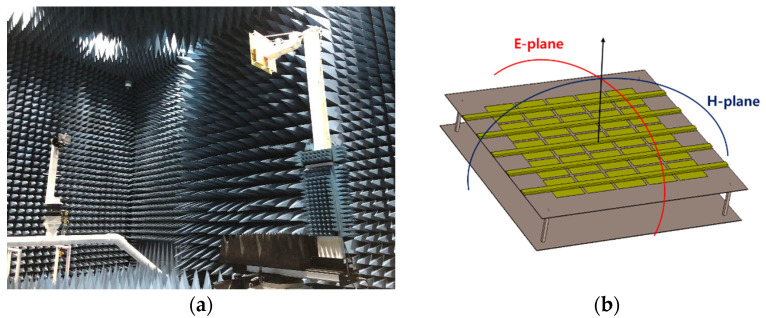
(**a**) Antenna radiation pattern and efficiency measurement environment; (**b**) measured plane.

**Figure 11 sensors-21-06508-f011:**
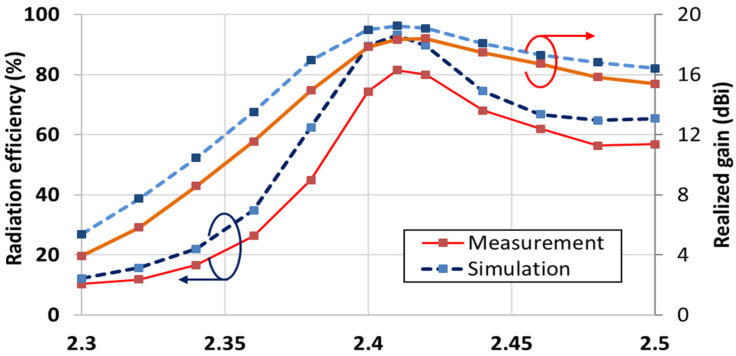
Antenna efficiency and gain comparisons according to frequency.

**Figure 12 sensors-21-06508-f012:**
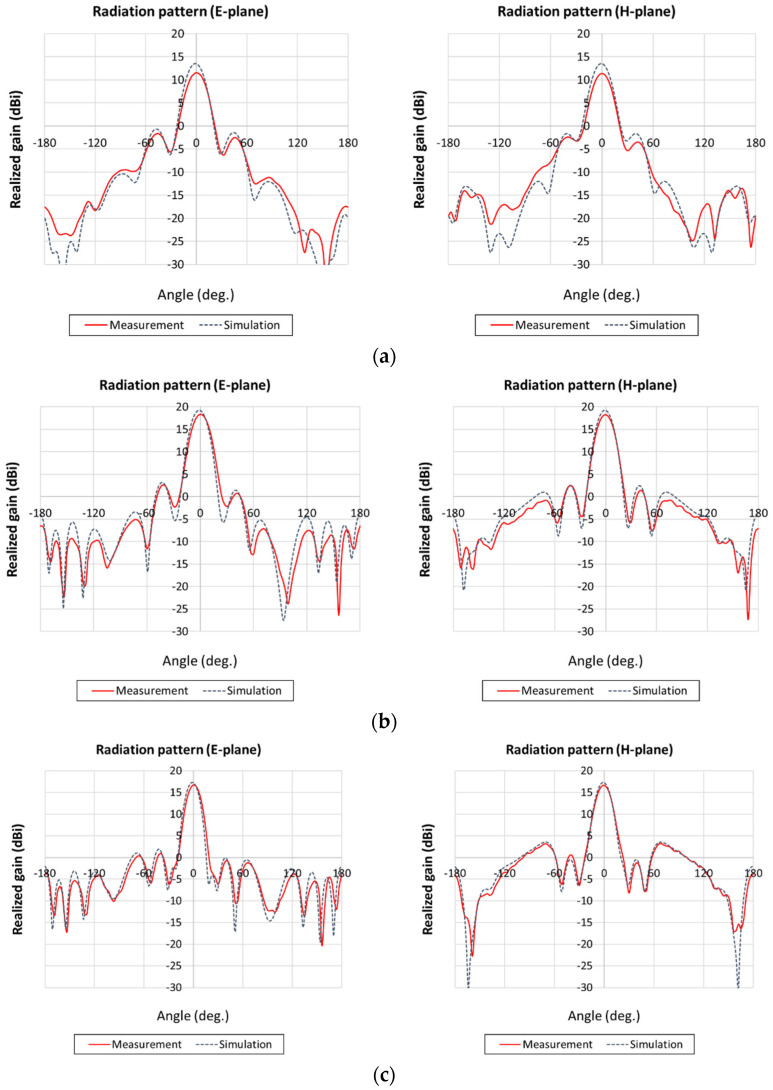
Simulated and measured radiation patterns of the antenna: (**a**) 2.36 GHz, (**b**) 2.41 GHz, (**c**) 2.46 GHz.

**Table 1 sensors-21-06508-t001:** The 3-dB beamwidth according to frequency.

Frequency		2.36 GHz	2.41 GHz	2.46 GHz
Simulation	E-plane	20.7°	17.8°	15.7°
	H-plane	20.8°	18.6°	17°
Measurement	E-plane	22°	18°	16°
	H-plane	21°	18°	17°

## Data Availability

Not applicable.

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
