# Peer review of "Multifunctional Partially Reflective Surface for Smart Blocks"

_sensors, 2021, doi:10.3390/s21196508_

Round 1
Reviewer 1 Report
The paper presented a 2D leaky-wave antenna employed for a smart block. Although LEDs lines are mounted on top of the PRS, the antenna can function properly with high and narrow gain radiation around 2.4 GHz. The paper contains some good results in general, but some sentences are not clear and need to be rephrased in good English. Some modifications and clarification are needed.
- The abstract is not understandable in general, and the sentences are not properly written.
- The authors in the abstract stated that PRS services only to increase the antenna gain. This is not true in general since it can also increase the gain-bandwidth by properly modified the PRS having multiple metal-dielectric layers.
- In the introduction, Sentences must be reworded and reconnected in a better way.
- It is not clear why are the authors discussing the FPC antenna gain properties in the introduction? And what is the relation between the smart block and the gain?
- In the unit-cell design in Fig. 5, clarify what is the imaginary PRS? and the spacing between each layer in the simulation?
- What is the functionality of the antenna when used as a smart block? What is the role of the LEDs?
- How the system can be distinguished between multiple passengers passing through the smart blocks?
Author Response
The paper presented a 2D leaky-wave antenna employed for a smart block. Although LEDs lines are mounted on top of the PRS, the antenna can function properly with high and narrow gain radiation around 2.4 GHz. The paper contains some good results in general, but some sentences are not clear and need to be rephrased in good English. Some modifications and clarification are needed.
1. The abstract is not understandable in general, and the sentences are not properly written.
=> Thank you for the in-depth comments. As pointed out by the reviewer, the abstract has been revised.
2. The authors in the abstract stated that PRS services only to increase the antenna gain. This is not true in general since it can also increase the gain-bandwidth by properly modified the PRS having multiple metal-dielectric layers.
=> We agree with the reviewer’s opinion.
=> We modified the expression to increase only the gain in the abstract to the expression that it is mainly used to increase the gain.
3. In the introduction, Sentences must be reworded and reconnected in a better way.
=> Thanks for the valuable comments. The introduction part has been revised and written again.
4. It is not clear why are the authors discussing the FPC antenna gain properties in the introduction? And what is the relation between the smart block and the gain?
=> The proposed smart block is to collect a fee when a person passes over the smart block. Therefore, the smart block must use an antenna with high gain. I mentioned FPC in the introduction because FPC is one of the good solutions
5. In the unit-cell design in Fig. 5, clarify what is the imaginary PRS? and the spacing between each layer in the simulation?
=> Thank you for the valuable comments.
=> The imaginary PRS means a mirror image of the PRS.
=> We changed the name in the figure. And we added the distance value between the layers in the figure.
6. What is the functionality of the antenna when used as a smart block? What is the role of the LEDs?
=> The antenna of the smart block must have high gain, and the LED should be installed on the block. In that case, the antenna should not be affected by this LED. In this paper, even if there is an LED, the PRS is not affected and the PRS increases the antenna gain.
7. How the system can be distinguished between multiple passengers passing through the smart blocks?
=> As shown in Figure 1, the width of one smart block is wide for one person to pass through. When a person passes by, the LED on the floor turns on when the payment is completed. If multiple people pass by, LEDs will be lit on each of the multiple smart blocks. If the LED does not light up despite a person passing by, it can be regarded as a person who has not paid the bill.
Reviewer 2 Report
Comments to the Author:
The authors present the multifunctional partially reflective surface for smart blocks. The reviewer finds the work interesting; however, below are some comments for the authors to consider.
- Could the authors cite more fundamental literature on the smart block?
- Boundary conditions described in Section 2 and illustrated in Figure 4 are questionable. According to the manuscript from line #118 to line #133, it seems that the authors attempt to explain the concept of “polarization grid” or “wire grid polarizer”. However, the illustration in Figure 4 is incorrect. Periodic metallic lines would be the correct form. Please the authors clarify this issue.
- The height between the patch antenna and PRS should be described in detail. The height design formula for initial value and some references should be added.
- Is it feasible to model the LED lines by a metallic line in the simulation? Please the authors clarify this issue.
- In Figure 9 and Figure 10, the labels of the vertical axis should be “|S11| (dB)”, not the “S11 (dB)”. Same errors can also be found in the text.
- The authors report the 3-dB beamwidth of the proposed antenna; however, both of the 3-dB beamwidth in E- and H-plane should be provided. If they are the same, the authors should state this.
Author Response
Reviewer 2
The authors present the multifunctional partially reflective surface for smart blocks. The reviewer finds the work interesting; however, below are some comments for the authors to consider.
- Could the authors cite more fundamental literature on the smart block?
=> Thank you for your kind comments. We also made an effort to find more references about smart blocks. However, it is difficult to insert references because there is hardly any research on inserting IoT sensors into sidewalk blocks, and smart block research is currently underway on railroad facilities. We ask for your understanding.
2. Boundary conditions described in Section 2 and illustrated in Figure 4 are questionable. According to the manuscript from line #118 to line #133, it seems that the authors attempt to explain the concept of “polarization grid” or “wire grid polarizer”. However, the illustration in Figure 4 is incorrect. Periodic metallic lines would be the correct form. Please the authors clarify this issue.
=> Thank you for your valuable comments. The reviewer's point of view is correct. As a single conductor, it is difficult to clarify about transmission and reflection of electromagnetic waves. The figure has been modified with periodic metallic lines.
3. The height between the patch antenna and PRS should be described in detail. The height design formula for initial value and some references should be added.
=> In Figure 2 we added the spacing (60 mm) between the patch antenna and the PRS. The parameter to design a high-gain radiation pattern on the designed frequency is the distance between the PRS and ground. The distance between the patch antenna and the PRS affects the input impedance. The following paper has been added as a reference for this.
Goudarzi, A.; Honari, M. M.; Mirzavand, R., Resonant Cavity Antennas for 5G Communication Systems: A Review. Electronics. 2020, 9 (7).
Nguyen-Trong, N.; Tran, H.H.; Nguyen, T.K.; Abbosh, A.M. A compact wideband circular polarized Fabry-Perot antenna using resonance structure of thin dielectric slabs. IEEE Access 2018, 6, 56333–56339.
=> And we added the following to the section 2
“Based on the analytical model such as the transmission line model, the distance between the ground plane and the PRS is the distance which makes the phase sum of the reflected waves becomes zero, which is about half wavelength.”
4. Is it feasible to model the LED lines by a metallic line in the simulation? Please the authors clarify this issue.
=> Thanks for the good comments.
=> For the LED line, two metal lines (+) and (-) are connected to the LED. Therefore, LEDs can be designed with metal wires.
=> This LED line can be attached on the metal line of the PRS. In this case, the performance of the antenna is the same in simulation with and without LED. Therefore, even if an actual LED is attached on the metal line of the PRS, the performance of the antenna is not affected at all. (figures are attached)
5. In Figure 9 and Figure 10, the labels of the vertical axis should be “|S11| (dB)”, not the “S11 (dB)”. Same errors can also be found in the text.
=> Thanks for the good comments. The S11 has been modified as follows in the figures and sentences.
S11 -> |S11|
6. The authors report the 3-dB beamwidth of the proposed antenna; however, both of the 3-dB beamwidth in E- and H-plane should be provided. If they are the same, the authors should state this.
=> Thanks for the valuable comments. We insert the 3dB beamwidth for the E- and H-plane in Table 1.

Reviewer 3 Report
The multifunctionality of the PRS has not been demonstrated in the manuscript. In fact, the presented results are similar to those of patch antennas with PRS superstrate that have been reported in numerous articles. There is no clear difference between the proposed design and the usual Fabry-Perot cavity configurations.
- The authors have correctly mentioned in the Introduction that Fabry-Perot cavities have the limitations of a narrow bandwidth. However, it is unclear if this problem has been addressed in the proposed design or not?
- Figure 1, is it safe for a passenger to pass through the main beam of a high gain antenna?
- Figure 2 presents a common knowledge and hence can be removed from the manuscript.
- Page 4, line 104, why a size of 40 cm for the substrate? This results in a physically large structure which may be impractical.
- Page 5, line 150: “Imaginary PRS” can be replaced by “a mirror image of the PRS”
- Page 5, line 162: Why a thickness of 20mm is considered?!
- Page 5, line 176, again a thickness of 10 mm is mentioned. Is this the thickness of metal lines, i.e. 1 cm? Is this practical? It is almost 0.8λ and will increase the size and cost significantly. However, from the diagram it appears as this is the width not the thickness, which is still significantly large compared to the effective wavelength.
Author Response
Reviewer 3
The multifunctionality of the PRS has not been demonstrated in the manuscript. In fact, the presented results are similar to those of patch antennas with PRS superstrate that have been reported in numerous articles. There is no clear difference between the proposed design and the usual Fabry-Perot cavity configurations.
=> Thanks for the good comments. The design of the square-shaped PRS is not much different from the existing papers. However, the purpose of this paper is to present a PRS structure that is not affected by the LED line all even if the LED line is attached to the PRS. For this purpose, the polarization characteristics of the antenna were used, and the width of the LED was made 10 mm and it was suggested that even if combined with the PRS, there is little effect on the performance of the antenna.
- The authors have correctly mentioned in the Introduction that Fabry-Perot cavities have the limitations of a narrow bandwidth. However, it is unclear if this problem has been addressed in the proposed design or not?
=> Thank you for your kind comments. In the case of single layer PRS, if the reflection is designed high, the bandwidth is narrowed. However, since there are many cases of using multi-layer PRS to solve this problem recently, the part about the narrowing bandwidth was judged to be incorrect and the sentence was removed.
- Figure 1, is it safe for a passenger to pass through the main beam of a high gain antenna?
=> The power is low because it is a communication signal for the cost charge. Therefore, it can be considered that there is no effect on the human body. The transmitter will transmit less than 1W and passengers will also pass by at the moment.
- Figure 2 presents a common knowledge and hence can be removed from the manuscript.
=> Thank you for your kind comments. Figure 2 is deleted.
- Page 4, line 104, why a size of 40 cm for the substrate? This results in a physically large structure which may be impractical.
=> 40 cm was selected considering the size of the sidewalk block. We had to redesign the sidewalk block for cost paymement, and it was judged to be an appropriate size considering the antenna gain.
- Page 5, line 150: “Imaginary PRS” can be replaced by “a mirror image of the PRS”
=> Thanks for the good comments. The name has been modified in the figure.
- Page 5, line 162: Why a thickness of 20mm is considered?!
=> First of all, the thickness of the LED line was modified with the width. I made a type error in the paper. Actual LED lines have various widths. In order to attach these LED lines to the PRS, I thought that a situation where the width would be 20 mm could occur. The metal line described as the LED line of the PRS is structured so that the actual LED line can be attached.
- Page 5, line 176, again a thickness of 10 mm is mentioned. Is this the thickness of metal lines, i.e. 1 cm? Is this practical? It is almost 0.8λ and will increase the size and cost significantly. However, from the diagram it appears as this is the width not the thickness, which is still significantly large compared to the effective wavelength.
=> the word of the paper was written incorrectly. I have modified the line width as shown below.
=> Thickness -> width
Round 2
Reviewer 3 Report
The authors have addressed all my comments.